# Adjuvant Effects of a New Saponin Analog VSA-1 on Enhancing Homologous and Heterosubtypic Protection by Influenza Virus Vaccination

**DOI:** 10.3390/vaccines10091383

**Published:** 2022-08-24

**Authors:** Noopur Bhatnagar, Ki-Hye Kim, Jeeva Subbiah, Bo Ryoung Park, Pengfei Wang, Harvinder Singh Gill, Bao-Zhong Wang, Sang-Moo Kang

**Affiliations:** 1Center for Inflammation, Immunity and Infection, Institute for Biomedical Sciences, Georgia State University, Atlanta, GA 30302, USA; 2Influenza Division, National Center for Immunization and Respiratory Diseases, Centers for Disease Control and Prevention, Atlanta, GA 30329, USA; 3Department of Chemistry, The University of Alabama at Birmingham, Birmingham, AL 35294, USA; 4Department of Chemical Engineering, Texas Tech University, Lubbock, TX 79409, USA

**Keywords:** influenza virus, split vaccine, adjuvant, cross-protection

## Abstract

Adjuvants can increase the magnitude and durability of the immune response generated by the vaccine antigen. Aluminum salts (Alum) remain the main adjuvant licensed for human use. A few new adjuvants have been licensed for use in human vaccines since the 1990s. QS-21, a mixture of saponin compounds, was included in the AS01-adjuvanted Shingrix vaccine. Here, we investigated the adjuvant effects of VSA-1, a newly developed semisynthetic analog of QS-21, on promoting protection in mice after vaccination with the inactivated split virus vaccine. The adjuvant effects of VSA-1 on improving vaccine efficacy after prime immunization were evident as shown by significantly higher levels of hemagglutination-inhibiting antibody titers and enhanced homologous protection compared to those by QS-21 and Alum adjuvants. The adjuvant effects of VSA-1 on enhancing heterosubtypic protection after two doses of adjuvanted vaccination were comparable to those of QS-21. T cell immunity played an important role in conferring cross-protection by VSA-1-adjuvanted vaccination. Overall, the findings in this study suggest that VSA-1 exhibits desirable adjuvant properties and a unique pattern of innate and adaptive immune responses, contributing to improved homologous and heterosubtypic protection by inactivated split influenza vaccination in mice.

## 1. Introduction

Influenza viruses infect between 5% and 15% of the global population, causing frequent hospitalization and up to 650,000 deaths annually [1]. Seasonal influenza vaccines can provide an effective intervention to limit the spread of the influenza virus. However, continual influenza virus mutation and its ability to evade immunity pose a constant threat. Annual vaccination is sub-optimally effective in conferring cross-protection against antigenically different influenza viruses. Hemagglutinin (HA), a major antigenic target of vaccination, contains the HA1 immunodominant variable head domain and the relatively conserved HA2 stalk domain being a target of cross-protective immunity [2]. It is of high priority to improve the cross-protective efficacy of influenza vaccines. Effective vaccination for increased efficacy of homologous and cross-protection could be achieved by using potent adjuvants that can be added along with a vaccine antigen to boost immune responses.

Adjuvants have become an indispensable component of successful subunit vaccines [3,4,5,6,7,8]. Adjuvants (a) enhance the ability of a vaccine to elicit strong and durable immune responses, in immunologically compromised individuals, such as immunologically immature neonates, the aged, and immune-suppressed individuals [9]; (b) reduce antigen dose and the number of immunizations; and (c) modulate the nature of the immune response. With the increasingly important role of adjuvants, several new adjuvants have been approved for human use [10,11,12,13]. Nonetheless, aluminum salts (Alum) are the most widely used adjuvants for licensed vaccine products. However, Alum has been shown to exhibit relatively weaker adjuvant effects than other adjuvants. Alum is efficient in eliciting antibody responses with a T helper (Th) type 2 profile but with limited efficacy [14].

QS-21, a key component of GlaxoSmithKline’s adjuvant system (AS) series included in the FDA-approved AS01b, might be one of the potent immunostimulants, which can promote the induction of strong and balanced humoral and cellular immune responses in combination with monophosphoryl lipid A (MPL) [15,16,17,18]. QS-21 is a mixture of two isomeric bidesmosidic saponins isolated from the tree bark of *Quillaja saponaria* Molina (QS), an evergreen tree native to temperate central Chile. However, the current shortage of QS-21, along with its dose-limiting toxicity, laborious and low-yielding purification, and chemical instability, limits its application and highlights the imperative need for its more accessible alternatives [19,20,21].

Extensive structure–activity relationship (SAR) studies of QS-21 led to the recent discovery of promising new saponin adjuvants that can be derived from more accessible natural sources [22,23,24,25,26]. Semisynthetic saponin adjuvant VSA-1 was prepared in one step from naturally occurring *Momordica* Saponin I [26], which was easily isolated from the inexpensive seeds of *Momordica cochinchinensis* SPRENG (MC), a widely available perennial vine [27]. VSA-1 saponin analog was reported to exhibit significant adjuvant activity [26]. Additionally, VSA-1 was shown to potentiate antigen-specific IgG1 and IgG2a immune responses in BALB/c mice, indicating a mixed Th1/Th2 immune response. VSA-1 showed lower acute toxicity than Quil A, which is a QS-21-like saponin adjuvant [26].

In this study, we investigated the adjuvant effects of VSA-1 on promoting homologous and heterosubtypic protection after prime or prime-boost vaccination of C57BL/6 mice with inactivated split 2009 influenza H1N1 pandemic virus vaccine. VSA-1 was found to exhibit desirable adjuvant properties by promoting humoral and cellular adaptive immune responses and contributing to improved homologous and heterosubtypic protection by inactivated split influenza vaccination in mice.

## 2. Materials and Methods

### 2.1. Animals, Reagents, and Viruses

Adult C57BL/6 and BALB/c mice (6- to 8-week-old, female) were purchased from Jackson Laboratory (Bar Harbor, ME, USA) and maintained in the animal facility at Georgia State University. All mouse studies were approved by Georgia State University Institutional Animal Care and Use Committee (IACUC, A21004) and carried out in compliance with the Guide for the Care and Use of Laboratory Animals of the NIH.

Preparation of VSA-1 was described in detail previously [26], where VSA-1 is a modified derivative of Momordica Saponin I isolated from the inexpensive seeds of *Momordica cochinchinensis* (Lour.) (Figure 1) [27]. VSA-1 was prepared as a white solid and then dissolved in pure autoclaved distilled water. Pure *Quillaja* saponin, QS-21, was purchased from Desert King International (San Diego, CA, USA) and was dissolved in dimethyl sulfoxide (DMSO) following the manufacturer’s protocol. VSA-1 and QS-21 were aliquoted and stored at −80 °C until use. Aluminum hydroxide (Alum) was purchased from Sigma Aldrich (Catalog number: A8222, St. Louis, MO, USA) and stored at 4 °C until use.

The split vaccine and homologous challenge virus strain used was A/California/04/2009 (A/Cal) H1N1. The heterologous challenge virus strain was reverse genetics (rg) reassortant H5N1 (rgH5N1), which contains HA and NA derived from A/Vietnam/1203/2004 and six internal genes from A/Puerto Rico/8/1934, as described previously [28]. Embryonated chicken eggs were used for propagating A/Cal H1N1 and rgH5N1 viruses. Inactivated viruses were prepared as described previously [29,30].

### 2.2. Immunization and Virus Challenge of Mice

The recommended dose of QS-21 adjuvant that can exhibit significant adjuvant effects in mice ranges from 5 µg to 20 µg and is typically 10 µg. The dose range for VSA-1 in mice is 25 µg to 2000 µg and is typically 50 µg or 100 µg. Although VSA-1 and QS-21 are saponin molecules, their molecular properties and tolerable dose ranges are different. To determine the protective efficacy of VSA-1 against homologous virus (A/Cal H1N1), groups of 6-week-old C57BL/6 mice (*n* = 6–8 per group) were intramuscularly immunized with sCal (3 μg) alone or sCal (3 μg) plus VSA-1 (50 μg for prime, 25 μg for boost), QS-21 (10 μg), or Alum (50 μg). Blood samples were collected at 2 weeks after immunization. The mice were challenged with a lethal dose of homologous A/Cal H1N1 virus (3 × LD_50_, equivalent to 5 × 10^3^ EID_50_) at 4 weeks after prime immunization. For cross-protection studies, C57BL/6 mice (*n* = 7 per group) were intramuscularly immunized (prime and boost) with sCal (3 μg) alone or sCal (3 μg) plus VSA-1 (50 μg prime; 25 μg boost), QS-21 (10 μg), or Alum (50 μg) at a 3-week interval. The mice were challenged with a lethal dose of heterosubtypic rgH5N1 virus (3 × LD_50_, equivalent to 2.5 × 10^3^ EID_50_) at 3 to 4 weeks after boost. After challenge, body weight changes and survival rates were monitored for 14 days for *n* = 3–5 per group. Lung viral titers and immunological profiles were determined in bronchoalveolar lavage (BAL), mediastinal lymph nodes (MLN), and lung and spleen tissues collected on day 5 (homologous challenge with A/Cal H1N1 virus) or day 6 (heterosubtypic challenge with rgH5N1 virus) post infection for *n* = 3 per group.

### 2.3. Antibody Enzyme-Linked Immunosorbent Assay (ELISA)

Antigen-specific antibody levels were measured by coating inactivated A/Cal H1N1 or rgH5N1 viruses (200 ng/well) onto ELISA plates and then incubating with diluted immune sera as detailed previously [29,31]. IgG isotypes were measured using horse-radish peroxidase (HRP)-conjugated anti-mouse immunoglobulin IgG, IgG1 and IgG2c secondary antibodies (Southern Biotechnology, Birmingham, AL, USA), and tetramethylbenzidine (TMB) substrate (Invitrogen, Waltham, MA, USA). Additionally, the consensus group I hemagglutinin (HA) stalk protein (50 ng/well), prepared as described [32], and N1 neuraminidase (NA) protein (20 ng/well, BEI Resources, NR-19234) were used to determine HA stalk and NA-specific IgG antibodies, respectively.

### 2.4. Hemagglutination Inhibition (HAI) Assay

To determine HAI titers in immune sera, the serum samples were treated with receptor destroying enzyme (RDE, Sigma-Aldrich, St. Louis, MO, USA), followed by inactivating (56 °C, 30 min) and mixing with an equal volume of 4 HA units of A/Cal H1N1 or rgH5N1 virus. HAI titers were determined as the highest dilution factor inhibiting the formation of buttons with 0.5% chicken red blood cells (RBC, Lampire Biological Laboratories, Pipersville, VA, USA) as previously described [29].

### 2.5. Lung Viral Titration

Lung extracts prepared in 1.5 mL of Roswell Park Memorial Institute (RPMI) 1640 by mechanical grinding of lung tissues harvested at day 5 or 6 after challenge were used to determine viral titers in embryonated chicken eggs (Hy-Line North America, LLC., Mansfield, GA, USA), as described previously [33]. Virus titers as 50% egg infection dose (EID_50_)/mL were evaluated according to the Reed and Muench method [34].

### 2.6. Cytokine ELISA and In Vitro IgG Antibody Detection

The levels of inflammatory cytokine tumor necrosis factor (TNF)-α from bronchoalveolar lavage fluids (BALF) and lung extracts were measured by cytokine ELISA using the Ready-SET-Go kit with TNF-α specific antibodies (eBioscience, San Diego, CA, USA) as described previously [35].

BALF was obtained by infusing 1.5 mL of PBS into the lungs. Lung extracts were prepared in 1.5 mL of RPMI 1640 by mechanical grinding of lung tissues harvested on day 5 after the challenge. BALF and lung extracts were used to determine vaccine antigen (A/Cal)-specific and rgH5N1-specific antibody levels.

Secreted IgG antibodies specific for A/Cal H1N1 or rgH5N1 were determined from MLN (5 × 10^5^ cells/well) and spleen tissues (5 × 10^5^ cells/well) from C57BL/6 mice. The cells from MLN and spleen were isolated at day 5 or 6 post infection and cultured for 1 day and 5 days in plates pre-coated with inactivated A/Cal H1N1 virus or inactivated rgH5N1 virus. The combined levels of IgG antibodies secreted into the culture supernatants and those captured on the plate were analyzed by ELISA.

### 2.7. Enzyme-Linked Immunospot (ELISpot) Assay

Interferon (IFN)-γ-secreting cell spots were determined by culturing splenocytes (5 × 10^5^ or 10^6^ cells/well) and lung cells (5 × 10^5^ or 3 × 10^5^ cells/well) for 72 h on multi-screen 96-well plates (MilliporeSigma, St. Louis, MO, USA) coated with IFN-γ capture antibody (BD Pharmingen) as described [28], in the presence of inactivated influenza A/Cal H1N1 (4 µg/mL) as an antigenic stimulator. The plates were then incubated with biotinylated mouse anti-IFN-γ antibody (BD Pharmingen), followed by incubation with alkaline phosphatase-labeled streptavidin antibody, and the IFN-γ-secreting T cells were visualized using color-developing 3,3′-diaminobenzidine substrate and counted using an ELISpot reader (BioSys, Miami, FL, USA).

### 2.8. Flow Cytometry Analysis

Lung and spleen tissues were harvested on day 5 or 6 after challenge. Lung cells harvested from the layer of percoll gradients between 44% and 67%, and spleen cells processed from spleen tissues were stimulated with 4 µg/mL inactivated A/Cal H1N1 virus in the presence of Brefeldin A (20 µg/mL) for 5 h at 37 °C as described [36,37]. In vitro cultured cells (lung and spleen cells) were stained with anti-CD3-PacificBlue (Clone 17A2, Biolegend, San Diego, CA, USA), anti-CD4-PE/Cy5 (Clone RM405, BD Biosciences, San Jose, CA, USA), and anti-CD8-FITC/Annexin V (Clone 53–6.7, eBioscience) antibodies, followed by fixation and permeabilization using BD Cyto-fix/CytopermTM Plus Kit (BD Biosciences). After staining the cells with anti-IFN-γ-APC/Cy7 (Clone XMG1.2, BD), anti-TNF-α-PE/Cy7 (Clone MP6-XT22, Biolegend), and anti-Granzyme B-FITC (Clone NGZB, eBioscience) antibodies, lymphocytes were first gated by forward versus side scatter strategic gating, followed by gating CD3^+^ T cells and then CD4^+^ T cells and CD8^+^ T cells secreting cytokines (Appendix A). The numbers of effector T cells in BAL and lung were expressed by reflecting the frequency gated out of the total cells from each mouse. Cells positive for intracellular cytokines were revealed through acquisition on a Becton-Dickinson LSR-II/Fortessa flow cytometer (BD, San Diego, CA, USA) and analyzed by Flowjo software (Tree Star Inc., Ashland, OR, USA).

### 2.9. In Vivo Protection Efficacy Test of Immune Sera

Immune sera collected at two weeks after boost immunization were diluted 50 folds, heat-inactivated at 56 °C for 30 min, followed by mixing with the same volume of 2.5 × LD_50_ A/Cal H1N1 virus and incubating at room temperature for 30 min as described [38]. The mixture of A/Cal H1N1 virus and sera was intranasally administered to naïve BALB/c mice (*n* = 3 per group), and body weight changes and survival rates were monitored daily for 14 days.

### 2.10. In Vivo Depletion of T Cells

For in vivo systemic T cell depletion before and post challenge, prime-boost immunized C57BL/6 mice (*n* = 3 per group) received treatment with anti-CD4 (CD4 clone GK1.5) and anti-CD8 (CD8 clone 53.6.7) monoclonal antibodies (mAbs) as described previously [39]. Antibodies (BioXCell, West Lebanon, NH, USA) were injected into the mice with intraperitoneal (IP; 1 day before challenge) and intranasal (IN; 1 day after challenge) sequential delivery at a 2-day interval (anti-CD4 200 μg and anti-CD8 150 μg/mouse for IP injection, 10 μg anti-CD4/8/mouse for IN inoculation). All groups (*n* = 3 per group) were challenged with a lethal dose of rgH5N1 influenza virus (3 × LD_50_), and body weight changes and survival rates were monitored daily for 14 days after challenge.

### 2.11. Intraperitoneal Injection of Adjuvants

Naïve C57BL/6 mice (*n* = 3 per group) were intraperitoneally injected with 200 μL of PBS, VSA-1 (50 µg), QS-21 (10 µg), or Alum (50 µg). The inflammatory cytokine, interleukin (IL)-6, and chemokines, namely keratinocytes-derived chemokine (KC) and monocyte chemoattractant protein 1 (MCP-1), were measured in blood samples collected at 2 h and 20 h after injection by cytokine and chemokine ELISA using Ready-SET-Go kits (eBioscience, San Diego, CA, USA). Cellular phenotypes in peritoneal exudates collected in 2 mL of PBS at 20 h after injection of adjuvants were determined by flow cytometry using cell-specific phenotypic markers as described [29].

### 2.12. Statistical Analyses

All results are presented as mean ± standard errors of the mean (SEM). The statistical significance was calculated by one-way or two-way analysis of variance (ANOVA). *p*-values ≤ 0.05 were considered significant. Data analysis was performed using Prism software (GraphPad Software Inc., San Diego, CA, USA).

## 3. Results

### 3.1. VSA-1 in Influenza Vaccination Exhibits Adjuvant Effects on Increasing the Magnitude of Virus-Specific IgG Antibodies and HAI Titers after a Single Dose

The recommended dose of QS-21 in mice is no more than 20 μg due to its dose-limiting toxicity, and its typical dose is 10 μg [40]. A previous study reported that VSA-1 exhibited much lower acute toxicity than the natural QS saponins, and mice could tolerate up to 2000 μg of VSA-1 [26]. Thus, in this study, we used VSA-1 at 25 μg to 50 μg dose.

To determine whether VSA-1 adjuvant would improve the efficacy of a single-dose influenza vaccination, groups of C57BL/6 mice were primed with sCal vaccine (3 µg/mouse) alone or along with the comparing adjuvants: VSA-1, QS-21, or Alum. VSA-1 only (without vaccine) was used as a mock control (Figure 2A). It was observed that the sCal + VSA-1 and sCal + QS-21 groups induced higher levels of virus-specific IgG after prime immunization than the sCal + Alum and vaccine-only groups (Figure 2B). The sCal + VSA-1 group also induced higher levels of virus-specific IgG1 and IgG2b than QS-21-adjuvanted influenza vaccination. Similar levels of IgG2c isotype antibody were induced by VSA-1- or QS-21-adjuvanted vaccination. In contrast, Alum-adjuvanted vaccination-induced an IgG1-dominant isotype and the lowest levels of IgG2c antibodies (Figure 2C–E). The adjuvanted sCal vaccine groups also induced higher levels of HA stalk-specific IgG antibodies (Figure 2F).

Both the VSA-1- and QS-21-adjuvanted groups showed higher levels of A/Cal H1N1-specific HAI titers than the Alum-adjuvanted and vaccine-only groups. The sCal + VSA-1 group showed a 2-fold higher titer than the sCal + QS-21 group (Figure 2G). These data suggest that the inclusion of a new adjuvant VSA-1 in influenza vaccination can be more effective in enhancing IgG1 and IgG2b isotypes and HAI titers compared to QS-21 and IgG2c isotype antibodies compared to Alum-adjuvanted vaccination.

### 3.2. Single Dose of VSA-1-Adjuvanted Influenza Vaccination Induces Enhanced Protection against Homologous A/Cal H1N1 Virus

To determine the adjuvant effects on improving homologous protection, the immunized C57BL/6 mice were challenged with A/Cal H1N1 virus at a lethal dose at 3 weeks after prime immunization. Body weight changes were monitored for 14 days (Figure 3A). By day 6–7 post challenge, mice in the sCal vaccine-only group displayed severe weight loss (over 20%) and died of infection similar to those observed in the naïve or adjuvant-only mock control groups. Additionally, the QS-21- and Alum-adjuvanted vaccine groups exhibited substantial weight loss (~18–20%) with 80% survival rates. Notably, the sCal + VSA-1 group displayed the least weight loss (~10%) among the adjuvanted groups with 100% survival rates and quickly recovering to normal weight (Figure 3B,C). Lung samples were collected on day 5 after challenge to determine virus titers in embryonated chicken eggs (Figure 3D). The sCal + VSA-1 and sCal + QS-21 groups showed significantly lower levels of virus titers (~100-fold difference) than those in the Alum-adjuvanted, sCal vaccine-only, and mock control groups. These data suggest that VSA-1 would be more effective than QS-21 and Alum adjuvants in enhancing the influenza vaccine efficacy of adjuvanted prime vaccination.

The levels of inflammatory cytokines provide an additional barometer for assessing the protective efficacy by preventing inflammation. The naïve infection control group showed the highest levels of TNF-α, IL-6, IFN-γ, and IL-1β cytokines in airway BALF and lung samples at 5 days after infection (Appendix A). In contrast, the sCal + VSA-1 group more effectively prevented the induction of inflammatory cytokine TNF-α than other adjuvanted groups (Figure 3E,F). Adjuvanted vaccine groups showed lower levels of inflammatory cytokines (IL-6, IFN-γ, IL-1β) compared to the vaccine-only group or naïve mice after lethal infection (Appendix A).

### 3.3. VSA-1-Adjuvanted Split Virus Vaccination Enhances Antibody-Secreting Cell and IFN-γ-Producing T Cell Responses

An important goal of vaccination is to induce long-lived antibody-secreting cell (ASC) responses. Vaccinated mice were challenged with A/Cal H1N1 virus after 3 weeks of prime vaccination, and spleen and MLN cells were harvested on day 5 after challenge for analysis of IgG antibodies secreted from ASCs in in vitro culture supernatants by ELISA (Figure 4A,B). The groups immunized with sCal vaccine-only and the split vaccine plus QS-21 or Alum showed low levels of IgG antibodies after a 1-day culture of spleen and MLN cells. Notably, the sCal+VSA-1 group showed significantly higher levels of IgG antibodies in cultures of spleen and MLN cells.

The VSA-1 and QS-21-adjuvanted vaccine groups induced significantly higher levels of mucosal IgG antibodies in BALF and lung extracts than the Alum-adjuvanted and vaccine-only control groups (Figure 4C,D). Enhanced levels of HA stalk domain-specific IgG antibodies were observed in the VSA-1-, QS-21-, and Alum-adjuvanted sCal groups (Figure 4E). The VSA-1- and QS-21-adjuvanted sCal groups showed significantly enhanced levels of A/Cal H1N1-specific HAI titers by over 6 folds at day 5 after challenge (Figure 4F) compared to those before challenge (Figure 2G). The Alum-adjuvanted, vaccine-only, and mock control groups did not show increased HAI titers at day 5 after challenge. These data suggest that B cells can be effectively primed for rapid recall to generate virus-specific IgG responses in mucosal and systemic sites upon challenge, even after a single dose of VSA-1-adjuvanted vaccination.

To determine adjuvant effects on eliciting cellular immune responses, lung cells and splenocytes harvested on day 5 after infection were stimulated with inactivated A/Cal H1N1 virus. The number of IFN-γ producing cells in the VSA-1-adjuvanted vaccination group was higher than those in the Alum- and QS-21-adjuvanted vaccination groups in spleen cells and higher than those in naïve infected mice in lung cells (Figure 4G,H). Intracellular cytokine staining (flow cytometry) data showed that IFN-γ^+^ CD4^+^ T and IFN-γ^+^ CD8^+^ T cells from the lungs were induced at higher levels in the VSA-1-adjuvanted sCal-vaccinated C57BL/6 mice than those in non-adjuvanted sCal and no vaccine (VSA-1 mock and naïve infection) groups (Figure 4I,J).

### 3.4. Boost Immunization with VSA-1-Adjuvanted Influenza Vaccine Further Enhances Virus-Specific IgG Antibodies, HAI Titers, and Homologous Protection

To determine the immune-boosting effects of VSA-1 adjuvant, groups of C57BL/6 mice were primed and then boosted with sCal vaccine (3 ug/mouse) alone or together with adjuvants. VSA-1 only was used as a mock control (Figure 5A). It was observed that the sCal + VSA-1 and sCal + QS-21 groups induced higher levels of virus-specific IgG, IgG1, IgG2b, and IgG2c after boost immunization than the other groups (Figure 5B, Appendix A). IgG antibodies specific for NA (Figure 5C) were induced at the highest levels in the VSA-1- and QS-21-adjuvanted sCal boost vaccinations. HAI titers were induced at the highest level in the VSA-1-adjuvanted sCal group, followed by the QS-21, and then the Alum, and vaccine-only groups at 2 weeks after boost (Figure 5D). The aim of in vivo efficacy test is to determine the correlative role of adjuvanted-vaccine induced antisera in conferring homologous protection (Figure 5E,F). Towards this aim, naïve BALB/c mice were intranasally inoculated with a mixture of A/Cal H1N1 virus and antisera collected from VSA-1, QS-21, or Alum-adjuvanted sCal-immunized mice, or naïve mice. Naïve sera did not provide protection against A/Cal H1N1 virus, as evidenced by severe weight loss (>25%) and 0% survival rates in naïve mice (Figure 5E,F). In contrast, immune sera from VSA-1- and QS-21-adjuvanted sCal groups conferred protection in naïve mice without any weight loss and 100% survival rates; meanwhile, antisera from the Alum-adjuvanted group provided protection to naïve mice with more severe weight loss (~18%) (Figure 5E,F). These data suggest that boost immunization with split virus together with VSA-1 adjuvant induces higher levels of virus-specific IgG isotype antibodies and HAI titers, and protection against A/Cal virus, than sCal-only and Alum-adjuvanted vaccination.

### 3.5. VSA-1-Adjuvanted Prime-Boost Influenza Vaccination Induces Cross-Protection against Heterosubtypic rgH5N1 Virus

A regimen of vaccination and heterosubtypic challenge is presented (Figure 6A). IgG antibodies specific for rgH5N1 virus (Figure 6B) and HA stalk (Figure 6C) were induced at the highest levels in the VSA-1- and QS-21-adjuvanted sCal groups after boost. To determine the adjuvant effects of VSA-1 on improving heterosubtypic protection, the immunized C57BL/6 mice were challenged with rgH5N1 virus. By day 6–7 post challenge, the sCal vaccine-only group displayed ~20% weight loss similar to that observed in naïve or adjuvant-only treated and infected mice. Notably, the sCal + VSA-1 group showed the least weight loss (~5%) with a 100% survival rate and quickly recovered body weight by day 8 post challenge, compared to adjuvant-only and vaccine-only groups, that showed severe weight loss and did not survive the lethal challenge (Figure 6D,E). On day 6 after infection, the sCal + VSA-1 and sCal + QS-21 groups showed 100-fold lower virus titers than the naïve infection control group. The sCal + VSA-1 group showed ~10-fold lower virus titers than the sCal only group (Appendix A), but there was no statistical significance between these two groups. High levels of TNF-α, IL-6, IFN-γ, and IL-1β cytokines were induced in the lung samples from naïve mice (Appendix A) at 6 days after infection. In contrast, the adjuvanted (VSA-1, QS-21) sCal vaccine groups showed lower levels of inflammatory cytokines (TNF-α, IL-6, IFN-γ, IL-1β) in the lung compared to the naïve mice with infection. 

Overall, these data suggest that VSA-1- and QS-21-adjuvanted split virus vaccination confers higher survival rates with prevention of severe weight loss and lung inflammation upon the heterosubtypic virus challenge than split vaccine-only and Alum-adjuvanted vaccination.

### 3.6. VSA-1 Adjuvanted Vaccination Induces T Cell-Dependent Protection against Heterosubtypic rgH5N1 Virus

IFN-γ^+^ and TNF-α^+^-CD4^+^ T and CD8^+^ T cells from lung and spleen tissues were induced at higher levels in the VSA-1- and QS-21-adjuvanted sCal vaccinated C57BL/6 mice than those in sCal and naïve groups day 6 post challenge with rgH5N1 virus (Figure 7A–D). To investigate whether T cell immunity would contribute to cross-protection, the VSA-1-adjuvanted sCal prime-boost immunized mice were treated with antibodies depleting CD4 and CD8 T cells right before and after challenge with rgH5N1 virus (Figure 7E). Severe weight loss (>25%) with a 0% survival rate was observed in the VSA-1-adjuvanted sCal-vaccinated mice after CD4 and CD8 T cell depletion. In contrast, non-depleted VSA-1-adjuvanted sCal-vaccinated mice showed 100% protection against weight loss after rgH5N1 virus challenge (Figure 7E,F). These results suggest that protection against heterosubtypic rgH5N1 virus is dependent on T cells in the mice vaccinated with sCal + VSA-1.

### 3.7. VSA-1-Adjuvanted Prime-Boost Vaccination Enhances Antibody-Secreting Cell and IFN-γ-Producing T Cell Responses upon Virus Infection

Spleen and MLN cells were harvested on day 6 after rgH5N1 challenge for analysis of IgG antibodies secreted from ASCs in in vitro culture supernatants by ELISA (Figure 8A–D). The groups immunized with the split vaccine only showed low levels of A/Cal H1N1 and rgH5N1-specific IgG antibodies after a 1-day and 5-day culture of spleen and MLN cells, whereas the VSA-1- and QS-21-adjuvanted split virus vaccinated groups showed higher levels of antibodies in cultures of spleen and MLN cells. The VSA-1- and QS-21-adjuvanted vaccine groups induced significantly higher levels of A/Cal H1N1 and rgH5N1 virus-specific mucosal IgG antibodies in lung extracts than the control groups (Figure 8E,F). 

To determine the adjuvant effects of VSA-1 on eliciting cellular immune responses, lung cells and splenocytes harvested on day 6 after infection were stimulated with inactivated A/Cal H1N1 virus or inactivated rgH5N1 virus. The numbers of A/Cal H1N1-specific and rgH5N1-specific IFN-γ producing cell spots were found to be higher in the VSA-1- and QS-21-adjuvanted vaccination group than those in the sCal-only vaccination and naïve infection groups in lung and spleen cells (Appendix A).

### 3.8. Acute Innate Immune Responses Are Differentially Modulated by VSA-1 and Other Comparing Adjuvants, QS-21, and Alum

To evaluate the acute innate immune effects of VSA-1 on inducing cytokines, chemokines, and cell recruitment at the site of injection, we injected naïve C57BL/6 mice with PBS, VSA-1, QS-21, or Alum intraperitoneally and determined cytokines and chemokines in sera and cell phenotypes in the peritoneal cavity (PC). VSA-1 and QS-21 were more potent in acutely inducing cytokine (IL-6) and chemokines (KC, MCP-1) in sera compared to Alum. VSA-1 and QS-21 induced moderate levels of these cytokines and chemokines within 2 h after injection, whereas after 20 h, low to background levels of IL-6 and chemokines (KC, MCP-1) were detected in sera, particularly from the VSA-1 group, suggesting transient induction (Figure 9A–C).

The phenotypes of innate cells infiltrated into the peritoneal cavity at 20 h after injection with adjuvants were determined by flow cytometry (Figure 9D–K). VSA-1 injection recruited monocytes, neutrophils, eosinophils, and dendritic cell (DC) subsets—activated DCs (aDCs), CD11b^+^ DCs, and plasmacytoid DCs (pDCs) in the peritoneal cavity at high levels, which were similar to Alum, whereas QS-21 induced these cells at low levels at the site of injection. Overall, VSA-1 modulates acute immune responses of cytokines, chemokines, and innate immune cells at the site of injection in a unique pattern different from the other compared adjuvants.

## 4. Discussion

VSA-1 is a newly developed semisynthetic saponin adjuvant based on extensive structure–activity relationship studies of QS-21, which is widely used as an adjuvant component in licensed human vaccines. AS01 is a combination of immunostimulants QS-21 and MPL with liposomes, and AS02 is a combination of QS-21 and MPL with an oil-in-water emulsion [41]. AS01b is included in the human Shingles vaccine (Shingrix) and malaria vaccine (Mosquirix), and AS02 in developmental tuberculosis, melanoma, and malaria vaccines [11,42,43,44]. Alum is a gold standard adjuvant commonly used in licensed human vaccines. In this study, we investigated the adjuvant effects of VSA-1 on enhancing the immunogenicity and efficacy of sCal (2009 H1N1 pandemic virus) influenza vaccination in C57BL/6 mice compared with the effects of QS-21 and Alum adjuvants. VSA-1 adjuvant effects could be more potent in stimulating the induction of IgG1, IgG2a, and IgG2b isotypes as well as HAI functional antibodies, and homologous protection with single-dose sCal vaccination compared to those by QS-21- and Alum-adjuvanted sCal vaccinations. Particularly, treatment of VSA-1-adjuvanted sCal vaccinated mice with antibodies depleting CD4 and CD8 T cells resulted in low protective efficacy, suggesting an important role of T cells in conferring cross-protection. The main effects of VSA-1 adjuvant appear to enhance the magnitude of the overall responses to the vaccine and challenge virus antigens.

Alum adjuvant is biased in skewing the immune responses to induce IgG1 isotype (Th2 type) antibodies. Consistent with this, in our study, Alum-adjuvanted vaccination induced IgG1 isotype-dominant antibody responses and exhibited weak adjuvant effects on inducing HAI antibodies, correlating with high lung viral titers and low efficacy of protection after challenge. In contrast, the VSA-1 adjuvant was effective in promoting the induction of IgG1, IgG2b, and Ig2c isotype antibodies and HAI titers at the highest levels, suggesting that it can be developed as a more potent and suitable vaccine adjuvant than QS-21. The action mechanisms of VSA-1 remain unknown. Previous studies on QS-21 would provide some insights into the mechanisms of saponin carbohydrate adjuvants. One hypothesis is that QS-21 might facilitate vaccine antigen uptake by antigen-presenting cells by interacting with lectin receptors through carbohydrate domains, stimulating certain cytokines that activate T cell and B cell responses [45]. An alternative mechanism is the cholesterol-dependent endocytosis of vaccine antigens and QS-21 into dendritic cells [46]. The high affinity of QS-21 to membrane cholesterol may lead to pore formation by destabilizing the membrane structure and may facilitate the delivery of vaccine antigens into the cytosol of antigen-presenting cells for further processing into peptides for T cell activation. QS-21 is proposed to stimulate T cells via the mitogen-activated protein kinase through CD2 molecules, resulting in the production of Th1 cytokines [45]. Studies on QS-21 in mouse antigen-presenting cells reported that QS-21, in combination with TLR4 agonist MPL, activated NOD-like receptor P3 (NLRP3) inflammasome, a multi-protein complex, inducing the subsequent release of pro-inflammatory cytokines and potentially contributing to INF-γ-mediated Th1 responses [47]. Alum (at a dose of 100 µg) and QS-21 (at a dose of 5 µg) have been known to induce necrotic cell death, where QS-21 induced macrophage and dendritic cell death in a caspase-1-, ASC-, and NLRP3-independent manner [47]. This necrotic cell death mechanism of QS-21 might have led to a decrease in the total cell numbers observed in the peritoneal exudates 24 h after mice were intraperitoneally injected with QS-21 in this study (Figure 9D–K). To abrogate the potential cell lytic activity and toxicity of QS-21, a delivery platform of cholesterol-based liposomes was utilized in the AS01 formulation [41]. Since cholesterol-quenched QS-21 retained adjuvant potency as free QS-21, the linking of cell lytic activity and an immune-stimulant effect of QS-21 is questionable. There are several drawbacks inherent to QS-21 as a natural product, including chemical instability and heterogeneity, scarcity, and dose-limiting toxicity. Therefore, VSA-1, a semisynthetic compound, would be a more homogeneous and potent vaccine adjuvant as supported by this study, would be non-toxic even at high doses, and safer than QS-21 [26].

In AS01 formulation, MPL and QS-21 were shown to synergistically stimulate the production of immune mediators such as IFN-γ, IL-12, and IL-18, as well as recruit neutrophils and monocytes and activate natural killer and innate lymphoid cells [48,49,50]. VSA-1 was found to be a more effective stimulator for recruiting monocytes and diverse dendritic cell populations to the site of injection than QS-21. In contrast, QS-21 induced the production of IL-6 inflammatory cytokine and MCP-1 chemokine in sera within 2 h post injection transiently at higher levels than VSA-1. It remains to be determined whether a combination of MPL and VSA-1 will exhibit synergistic adjuvant effects as AS01 in future studies.

It is highly significant to enhance the effectiveness of seasonal vaccination by formulating vaccines containing safe adjuvants. Induction of T cell immunity and HA stalk and NA IgG antibodies was previously reported to be independently correlated with cross-protection [51]. Both VSA-1- and QS-21-adjuvanted sCal vaccinations induced cross-protection against rgH5N1 virus even in the absence of cross-reactive HAI antibodies. HA stalk-specific IgG antibodies, as well as possibly recall IFN-γ^+^ and TNF-α^+^ CD4 and CD8 T cell responses, were observed in the VSA-1- and QS-21-adjuvanted sCal groups at significantly higher levels, which appears to correlate with cross-protection. Particularly, the depletion of CD4 and CD8 T cells led to abrogation of cross-protection, supporting a critical role of cross-protective T cell immunity induced by VSA-1-adjuvanted vaccination.

In summary, VSA-1, as a new semisynthetic saponin-based adjuvant, played a significant role in enhancing IgG isotype and HAI functional antibodies after influenza split prime vaccination, conferring superior homologous protection over split vaccine, and Alum- and QS-21-adjuvanted vaccinations in C57BL/6 mice. Both VSA-1- and QS-21-adjuvanted sCal boost vaccinations were highly effective in inducing humoral and cellular immune responses as well as cross-protection against rgH5N1 virus, even in the absence of cross-reactive HAI antibodies. This study provides supportive evidence warranting further development of VSA-1 as a promising alternative candidate for QS-21 replacement.

## Figures and Tables

**Figure 1 vaccines-10-01383-f001:**
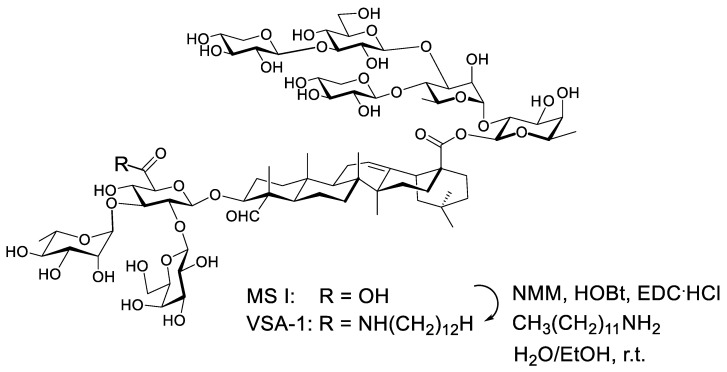
The structure of VSA-1 derived from Momordica saponin I, which was isolated from the seeds of *Momordica cochinchinensis*. Adapted with permission from Ref. [26]. Copyright © 2019, American Chemical Society.

**Figure 2 vaccines-10-01383-f002:**
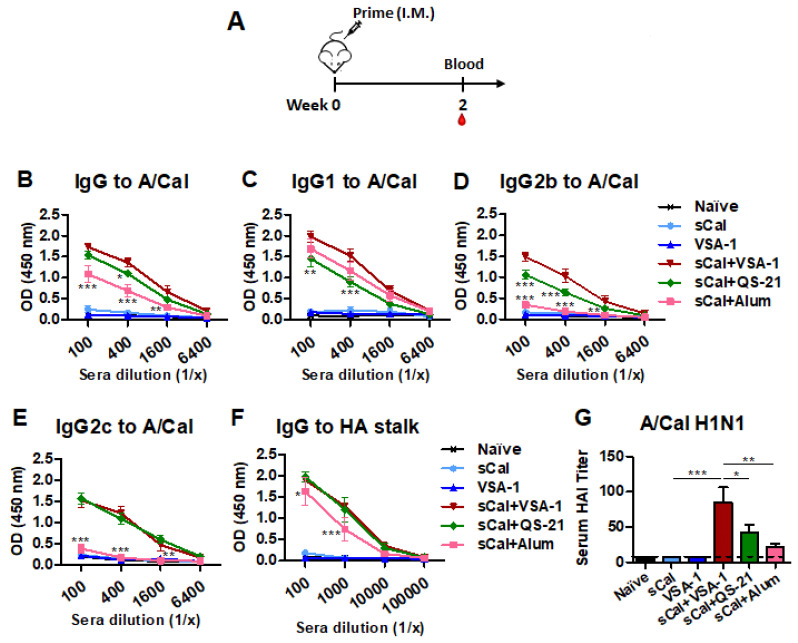
Prime immunization with VSA-1-adjuvanted split (sCal) vaccine enhances virus-specific IgG antibodies and HAI titers in C57BL/6 mice. (**A**) Immunization scheme (*n* = 5 per group). VSA-1: Mock (no vaccine) VSA-1 control, Naïve: PBS control. (**B**–**E**) A/Cal H1N1 virus-specific IgG and isotype antibody levels in prime sera. (**F**) IgG specific for consensus group I HA stalk. (**G**) HAI titers against A/Cal H1N1 virus. The dotted line represents the limit of detection. Error bars indicate the mean ± standard errors of the mean (SEM). *; *p* < 0.05, **; *p* < 0.01, ***; *p* < 0.001 compared to the sCal + VSA-1 group.

**Figure 3 vaccines-10-01383-f003:**
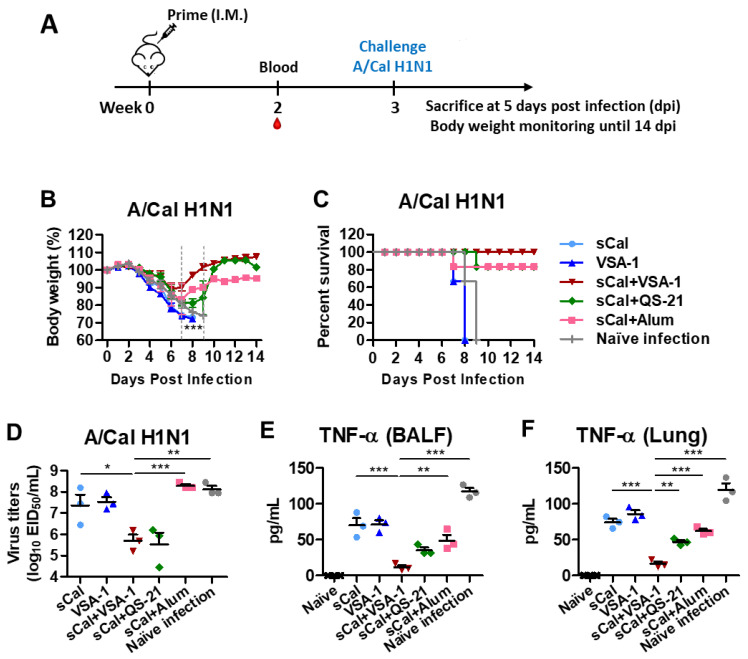
Prime immunization with VSA-1-adjuvanted sCal vaccine confers protection against lethal challenge with homologous A/Cal H1N1 virus. (**A**) Immunization and virus challenge scheme (*n* = 6–8 per group). (**B**) Body weight changes and (**C**) survival rates for 14 days (*n* = 3–5). (**D**) Lung viral titers (*n* = 3) as 50% egg infectious titers (EID_50_). (**E**,**F**) TNF-α cytokine levels in BALF and lung extracts. Error bars indicate the mean ± SEM. *; *p* < 0.05, **; *p* < 0.01, ***; *p* < 0.001 compared to the sCal + VSA-1 group.

**Figure 4 vaccines-10-01383-f004:**
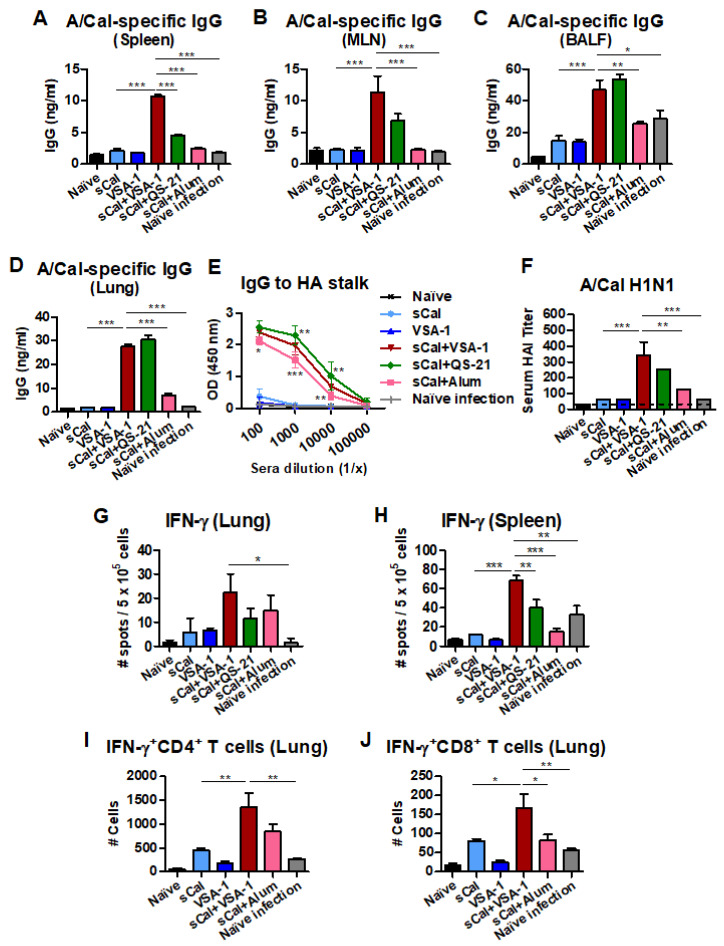
VSA-1-adjuvanted sCal vaccination enhances antibody-secreting cells and recall effector T cells on day 5 after A/Cal H1N1 influenza virus infection. (**A**,**B**) A/Cal virus-specific IgG (in vitro) production in splenocytes and MLN cells (*n* = 3 per group). (**C**,**D**) A/Cal virus-specific IgG levels in BALF and lung samples. (**E**) Antibody responses specific for HA2 stalk in sera after challenge. (**F**) HAI titers against A/Cal H1N1 virus. (**G**,**H**) Cytokine ELISpot of lung cells and splenocytes after in vitro stimulation with inactivated A/Cal virus. (**I**,**J**) Flow cytometry data of IFN-γ-secreting CD4^+^ and CD8^+^ T cells in lung samples. Error bars indicate the mean ± SEM. *; *p* < 0.05, **; *p* < 0.01, ***; *p* < 0.001 compared to sCal + VSA-1 group.

**Figure 5 vaccines-10-01383-f005:**
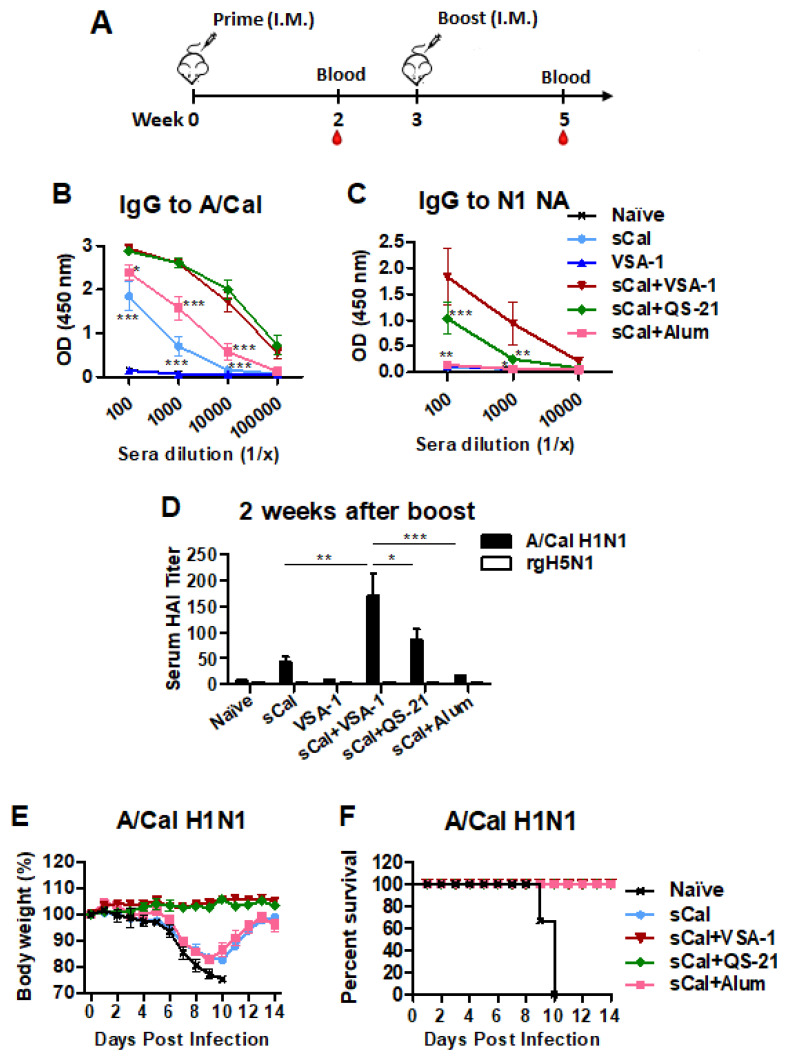
VSA-1-adjuvanted prime-boost sCal vaccination enhances IgG antibodies, HAI titers, and homologous protection. (**A**) Immunization scheme (*n* = 7). (**B**) A/Cal H1N1 virus-specific IgG and isotype antibody levels in boost sera. (**C**) N1 NA-specific IgG antibody levels in boost sera. (**D**) HAI titers against A/Cal H1N1 and rgH5N1 viruses in boost sera. (**E**,**F**) Body weights and survival rates to determine the role of boost antisera in protection. Error bars indicate the mean ± SEM. *; *p* < 0.05, **; *p* < 0.01, ***; *p* < 0.001 compared to the sCal + VSA-1 group.

**Figure 6 vaccines-10-01383-f006:**
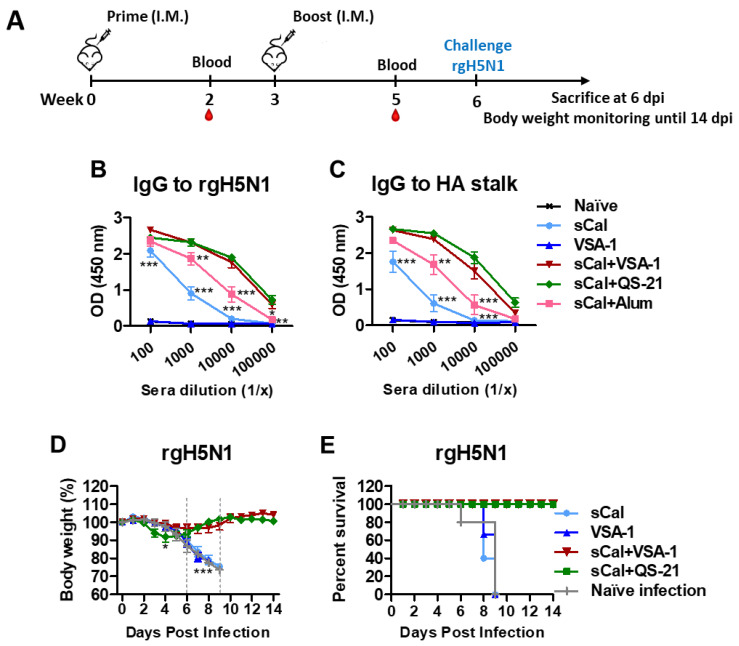
VSA-1-adjuvanted prime-boost sCal vaccination confers protection against lethal infection with heterosubtypic rgH5N1 virus. (**A**) Immunization and virus challenge scheme (*n* = 7). (**B**) rgH5N1 virus-specific IgG antibody levels in boost sera. (**C**) HA2 stalk-specific IgG antibody levels in boost sera. (**D**) Body weight changes and (**E**) survival rates after challenge with a lethal dose of rgH5N1 virus (3 × LD_50_) for *n* = 4 per group. Error bars indicate the mean ± SEM. *; *p* < 0.05, **; *p* < 0.01, ***; *p* < 0.001 compared to the sCal + VSA-1 group.

**Figure 7 vaccines-10-01383-f007:**
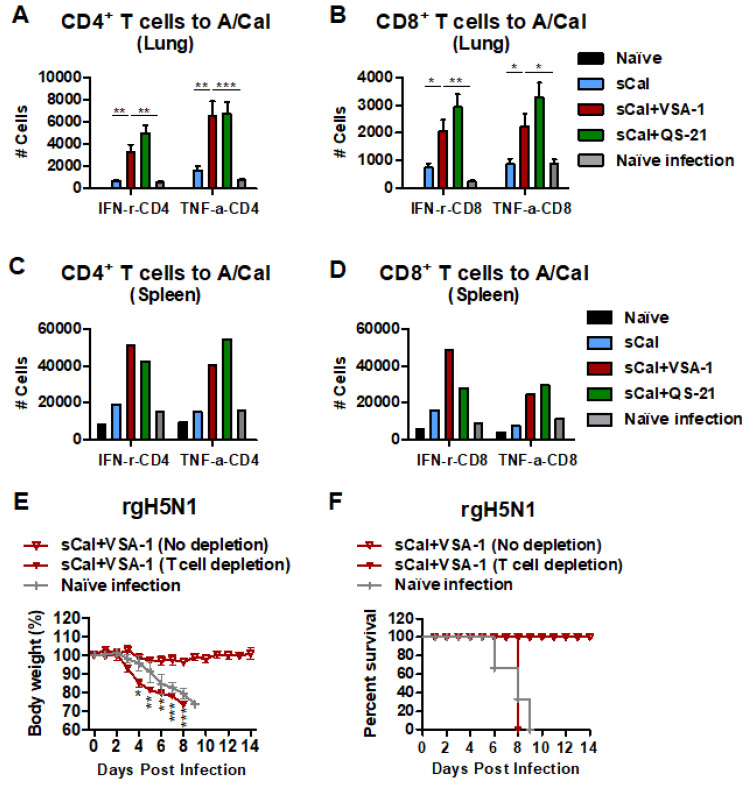
VSA-1-adjuvanted sCal vaccination confers T cell-mediated protection against rgH5N1 virus. (**A**–**D**) Flow cytometry data of IFN-γ-secreting CD4^+^ and CD8^+^ T cells in (**A**,**B**) lung and (**C**,**D**) spleen samples. (**E**) Body weight changes and (**F**) survival rates in VSA-1-adjuvanted prime-boost-immunized C57BL/6 mice (*n* = 3) with CD4 and CD8 T cells depleted and rgH5N1 virus challenge. Error bars indicate the mean ± SEM. *; *p* < 0.05, **; *p* < 0.01, ***; *p* < 0.001 compared to the sCal + VSA-1 group.

**Figure 8 vaccines-10-01383-f008:**
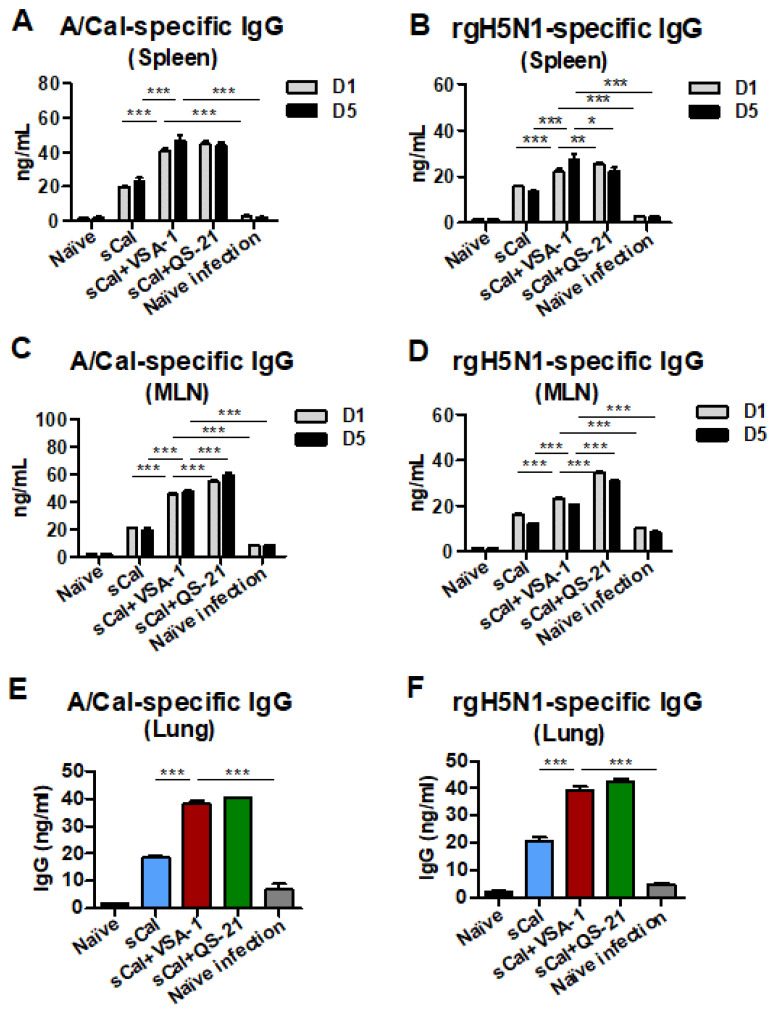
VSA-1-adjuvanted sCal vaccination enhances antibody-secreting cell responses. (**A**–**D**) A/Cal virus-specific and rgH5N1 virus-specific IgG (in vitro) production from (**A**,**B**) splenocytes and (**C**,**D**) MLN cells (*n* = 4). (**E**) A/Cal virus-specific and (**F**) rgH5N1 virus-specific IgG levels in lung samples. Error bars indicate the mean ± SEM. *; *p* < 0.05, **; *p* < 0.01, ***; *p* < 0.001 compared to the sCal + VSA-1 group.

**Figure 9 vaccines-10-01383-f009:**
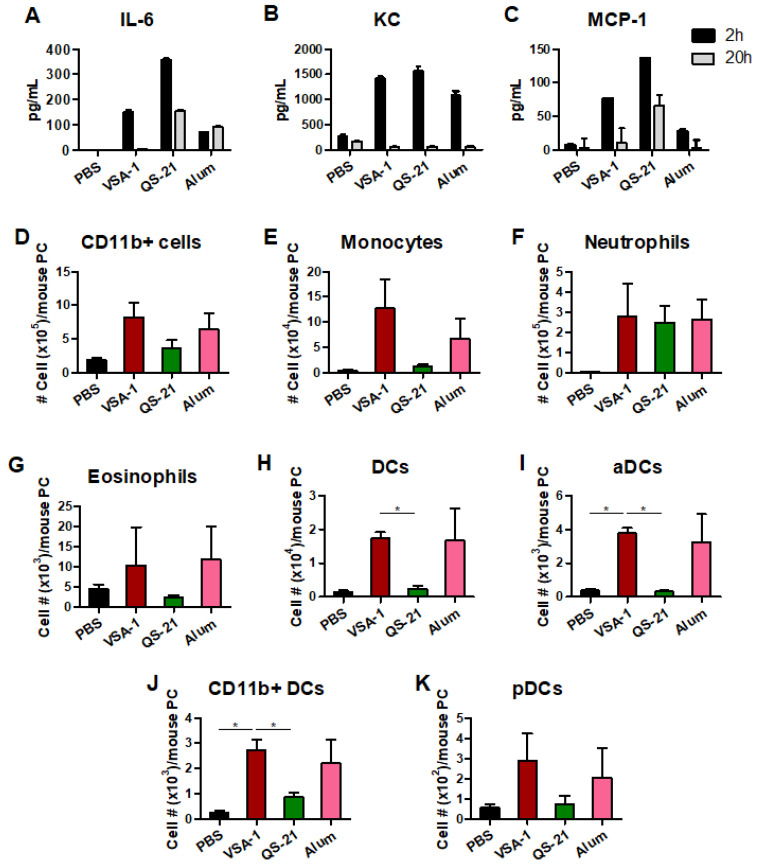
Acute innate immune induction of cytokines and chemokines in sera and innate cellular infiltrates in the peritoneal cavity (PC) after intraperitoneal injection of adjuvants. Naïve C57BL/6 mice (*n* = 3 per group) were intraperitoneally injected with PBS, VSA-1 (50 μg), QS-21 (10 μg), or Alum (50 μg). (**A**–**C**) Cytokine and chemokine ELISA levels in sera. (**D**–**K**) Cellular phenotypes in peritoneal exudates. Error bars indicate the mean ± SEM. *; *p* < 0.05 compared to the VSA-1 group.

## Data Availability

All data are available in the main text or Appendix A.

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
