# Peer review of "Adjuvant Effects of a New Saponin Analog VSA-1 on Enhancing Homologous and Heterosubtypic Protection by Influenza Virus Vaccination"

_vaccines, 2022, doi:10.3390/vaccines10091383_

Round 1

Reviewer 1 Report

Bhatnagar et al. test a newly described saponin like adjuvant, VSA-1, which is an easy to derive, synthetic version of widely used QS-21. Prior studies have indicated that VSA-1 might have lower acute toxicity than QS-21. The major strength of these studies are that the in vivo efficacy of VSA-1 as a vaccine adjuvant thus far has not been well described, and the text is convincing that VSA-1 can improve antibody responses, and likely T cell responses and protection with inactivated influenza antigen. The major weakness of the paper is that a number of claims are made that are not supported by the data presented. 

L13 and throughout the text: No data is provided that adjuvants affect the breadth of immune responses.

Authors need to flesh out the data interpretation. Increasing the breadth of an immune response in the most strict sense can occur by increasing the antigenic diversity of the immunogen. However, it can be that by increasing the magnitude of the response overall, or even somehow preferentially to sub-dominant or non-immunogenic epitopes, that heterosubtypic protection can be achieved. Additionally, adjuvants may affect somatic hypermutation and/or avidity maturation in B cell responses, as well as T cell receptor avidity. It is unclear from the text which of these or other processes the authors hypothesize to be occurring, and a short discussion of these and other mechanisms of increasing “breadth” of protection and/or immune responses by adjuvants is warranted given that is the central topic.

L35-36: “Annual vaccination is ineffective in conferring cross-protection against antigenically different influenza viruses.”

This statement would be better stated as that these vaccines are sub-optimally effective. As you stated above, seasonal vaccines can provide protection, and vaccine strains are often antigenically different to some degree to circulating strains.

L53 It would be controversial not “arguable” to say generically that QS-21 is “the most potent immunostimulant” and that it can induce “balanced cellular immune responses”. Please provide citations for the ability of QS-21 to elicit potent CD8+ T cell responses. The citations provided were studying a mixture of saponin and MPLA.

L77: It is not mentioned why only female mice were used for these studies.

L88-89: It is not clear if the aluminum hydroxide used here is of the precipitated form commonly used as an adjuvant, or if that it was endotoxin free. Please include a catalog number for the exact type used in these studies.

L110: Why was VSA-1 used at 50ug per mouse, but QS-21 used at 10ug per mouse? In lines 220-223 the justification is given that higher doses can be tolerated for VSA-1. Is there data or citable studies that directly compare the toxicity of these two adjuvants at these specific doses to show that it is an equivalent comparison? Conclusions must be carefully stated as an observed increase in immunogenicity may be due simply to increased dose. It is noted multiple times in the results that there was an increase in the magnitude of a particular immune response or enhanced protection elicited by VSA-1 compared with QS-21, but it is not addressed within the discussion that this could be purely due to using 5 times as much adjuvant. The citation (25) given in the discussion indicates that a 50ug dose for both may be well tolerated.

L168-184 It is not indicated if a viability stain was used in these studies, and if one was used, which one? It is said that “live lymphocytes were first gated”, but if this was simply a viability count done before cell stimulation, it would not be correct to imply that live cells were part of a gating strategy. There was no indication from the methods that an anti-CD8 antibody was used for flow cytometry analysis, yet there is data presented for CD8 T cell cytokine secretion analyzed by flow cytometry in figure 4 and others. Please include in the supplemental information the associated gating strategies and representative FACS plots for any flow cytometry data.

L198: Why was depletion of T cells by the intranasal route administered at a dose of only 10ug of depleting antibody? Was any analysis performed after depletion to determine depletion of CD8 and CD4 T cells occurred?

L214: Why was a correction for multiple comparisons used in statistical tests? It seems that the increase in power is unnecessary as the groups used in these studies are completely independent of each other? It seems that Fisher's Least Significant Difference (LSD) test would be the most appropriate here.

Figure 3 and associated text: it is unclear from the text how many mice were euthanized at D5 and how many were monitored for 14 days after challenge. If n=5 was used for both viral titers and weight loss data sets, please indicate that directly in the methods and all figure legends. It would also help if figures representing data as bars (eg. 3d-f) plot the data for individual samples.

L280-286: Is there histological data to augment the argument here of decreased pathology?

Figure 4I-J: How are total numbers of cells being calculated? Are the number of cells represented here for the entire lung, or a subsection of it? Typical preparation of single cell suspensions from lungs would yield >5 million cells per lung, which would indicate that in some animals the frequency of cells expressing IFNg is 1:10,000th of a percent. In the main text or supplemental text, the frequency of IFNg secreting cells as a fraction of CD4 or CD8 T cells should be reported, and representative FACS plots displayed.

L387: It is not accurate to say that vaccination suppressed the induction of inflammatory cytokines, only that there were lower levels. It could be that these cytokines are produced in response to higher levels of virus in the lung, and that vaccinated animals were protected from viral replication, and therefor had lower levels.

Figure 6D-E sCal+ alum data is not shown for these weight loss and viral titer protection graphs, yet it is stated in L389-391 that “These data suggest that VSA-1-adjuvanted split virus vaccination confers higher efficacy of protection against heterosubtypic virus than split virus only and Alum-adjuvanted vaccination.” Additionally, there was not a significant difference shown in the protection against viral lung titers between VSA-1+ sCal and unadjuvanted sCal. This is a major weakness of the paper, as a major claim is that this adjuvant enhances heterosubtypic protection, and this claim is not supported by the data. Further studies appear to be needed to support or reject this hypothesis, or the paper rewritten to limit conclusions.

L394 and throughout the text: Induced is not the correct term, as no data is presented that any of the vaccines induced a T cell response independent of viral infection. All T cell data in the text is obtained from samples collected after challenge. And while these likely could be recall responses, it is also very possible that these formulations simply accelerated a primary T cell response to infection. Contrary to claims made throughout the paper (eg. L481-482), there is no data to demonstrate that any vaccines elicited a T cell response. While protection was dependent on administration of T cell depleting antibody treatments, there is no data to show that T cells were depleted by antibody treatment.

Throughout the text, it is claimed that alum induced Th2 type responses, but Th2 cytokines were not measured.

Author Response

Responses to the comments from Reviewer 1

Bhatnagar et al. test a newly described saponin like adjuvant, VSA-1, which is an easy to derive, synthetic version of widely used QS-21. Prior studies have indicated that VSA-1 might have lower acute toxicity than QS-21. The major strength of these studies are that the in vivo efficacy of VSA-1 as a vaccine adjuvant thus far has not been well described, and the text is convincing that VSA-1 can improve antibody responses, and likely T cell responses and protection with inactivated influenza antigen. The major weakness of the paper is that a number of claims are made that are not supported by the data presented. 

Response: We appreciate for careful reading and providing constructive critiques to improve the clarity of the manuscript. We provide responses to all review comments and revised the manuscript accordingly.

L13 and throughout the text: No data is provided that adjuvants affect the breadth of immune responses.

Authors need to flesh out the data interpretation. Increasing the breadth of an immune response in the most strict sense can occur by increasing the antigenic diversity of the immunogen. However, it can be that by increasing the magnitude of the response overall, or even somehow preferentially to sub-dominant or non-immunogenic epitopes, that heterosubtypic protection can be achieved. Additionally, adjuvants may affect somatic hypermutation and/or avidity maturation in B cell responses, as well as T cell receptor avidity. It is unclear from the text which of these or other processes the authors hypothesize to be occurring, and a short discussion of these and other mechanisms of increasing “breadth” of protection and/or immune responses by adjuvants is warranted given that is the central topic.

Response: We are now very careful in the description of the experimental data and in the usage of the terms of immune responses. The main effects of VSA-1 adjuvant appear to enhance the magnitude of the overall responses to vaccine and challenge virus antigens but not the cross-neutralizing HAI antibodies. A brief discussion is provided about the main adjuvant effects in the inactivated influenza vaccination (page 19).

L35-36: “Annual vaccination is ineffective in conferring cross-protection against antigenically different influenza viruses.”

This statement would be better stated as that these vaccines are sub-optimally effective. As you stated above, seasonal vaccines can provide protection, and vaccine strains are often antigenically different to some degree to circulating strains.

Response: As suggested, this statement is revised. “Annual vaccination is sub-optimally effective in conferring cross-protection against antigenically different influenza viruses.”

L53 It would be controversial not “arguable” to say generically that QS-21 is “the most potent immunostimulant” and that it can induce “balanced cellular immune responses”. Please provide citations for the ability of QS-21 to elicit potent CD8+ T cell responses. The citations provided were studying a mixture of saponin and MPLA.

Response: This sentence is modified to deliver more general information: “QS-21, a key component of GlaxoSmithKline’s adjuvant system (AS) series, including the FDA-approved AS01b, might be one of potent immunostimulants, which can significantly increase the magnitude of both humoral and cellular immune responses to the vaccine antigens, in combination with monophosphoryl lipid A”.

L77: It is not mentioned why only female mice were used for these studies.

Response: There were no significant differences in IgG immune responses to vaccination between male and female mice (unpublished data). Technically, we had experiences of difficulty for social housing of male mice in a cage. There is no scientific reason for using female mice in this study.

L88-89: It is not clear if the aluminum hydroxide used here is of the precipitated form commonly used as an adjuvant, or if that it was endotoxin free. Please include a catalog number for the exact type used in these studies.

Response: The catalog number of the aluminum hydroxide used in this study is provided in the revised manuscript.

L110: Why was VSA-1 used at 50ug per mouse, but QS-21 used at 10ug per mouse? In lines 220-223 the justification is given that higher doses can be tolerated for VSA-1. Is there data or citable studies that directly compare the toxicity of these two adjuvants at these specific doses to show that it is an equivalent comparison? Conclusions must be carefully stated as an observed increase in immunogenicity may be due simply to increased dose. It is noted multiple times in the results that there was an increase in the magnitude of a particular immune response or enhanced protection elicited by VSA-1 compared with QS-21, but it is not addressed within the discussion that this could be purely due to using 5 times as much adjuvant. The citation (25) given in the discussion indicates that a 50ug dose for both may be well tolerated.

Response: The recommended dose range of QS-21 adjuvant effects in mice is 5-20 µg, and typically 10 µg. The dose range of VSA-1 in mice is 25 µg -2000 µg, and typically 50 µg or 100 ug. Although both are saponin molecules, their molecular properties and tolerable dose ranges are different. VSA-1 still has a wider range to increase its dose for stronger immune responses, on the other hand, QS-21 doesn't have much room to increase its dose without introducing side effects. QS-21 dose of 50 µg would be too toxic to mice and has not been tested. A brief description is included in the revised manuscript (page 3).

L168-184 It is not indicated if a viability stain was used in these studies, and if one was used, which one? It is said that “live lymphocytes were first gated”, but if this was simply a viability count done before cell stimulation, it would not be correct to imply that live cells were part of a gating strategy. There was no indication from the methods that an anti-CD8 antibody was used for flow cytometry analysis, yet there is data presented for CD8 T cell cytokine secretion analyzed by flow cytometry in figure 4 and others. Please include in the supplemental information the associated gating strategies and representative FACS plots for any flow cytometry data.

Response: A viability strain was not used in the flow cytometry experiments. We modified the description of this part, including anti-CD8 antibody staining. The addressed gating strategies and representative FACS plots are now presented in the supplemental information of the revised manuscript. (Supplementary Figure S1).

L198: Why was depletion of T cells by the intranasal route administered at a dose of only 10ug of depleting antibody? Was any analysis performed after depletion to determine depletion of CD8 and CD4 T cells occurred?

Response: The efficacy of depleting both CD4 and CD8 T cells by using the concentrations described was shown to be highly effective, over 99% as we reported (1). The IP injection was carried out with a regular dose (anti-CD4 200 μg and anti-CD8 150 μg /mouse), and then a low dose of IN inoculation was performed: “Antibodies (BioXCell, West Lebanon, NH) were injected into the mice with intraperitoneal (IP; 1 day before challenge) and intranasal (IN; 1 day after challenge) sequential delivery at a 2-day interval (anti-CD4 200 μg and anti-CD8 150 μg /mouse for IP injection, 10 μg anti-CD4/8/mouse for IN inoculation)”.

L214: Why was a correction for multiple comparisons used in statistical tests? It seems that the increase in power is unnecessary as the groups used in these studies are completely independent of each other? It seems that Fisher's Least Significant Difference (LSD) test would be the most appropriate here.

Response: We revised the stat analysis description, focusing on the methods only used in the data analysis of this study.

Figure 3 and associated text: it is unclear from the text how many mice were euthanized at D5 and how many were monitored for 14 days after challenge. If n=5 was used for both viral titers and weight loss data sets, please indicate that directly in the methods and all figure legends. It would also help if figures representing data as bars (eg. 3d-f) plot the data for individual samples.

Response: We provide the information on the numbers of mice for monitoring body weight changes for 14 days and for analysis post challenge at day 5 in the legend of the Figure 5. Also, the Figures 3D-F data were presented with the data for individual samples.

L280-286: Is there histological data to augment the argument here of decreased pathology?

Response: We do not have histological data.

Figure 4I-J: How are total numbers of cells being calculated? Are the number of cells represented here for the entire lung, or a subsection of it? Typical preparation of single cell suspensions from lungs would yield >5 million cells per lung, which would indicate that in some animals the frequency of cells expressing IFNg is 1:10,000th of a percent. In the main text or supplemental text, the frequency of IFNg secreting cells as a fraction of CD4 or CD8 T cells should be reported, and representative FACS plots displayed.

Response: In our preparation of lung single cell suspensions from lung tissues from a mouse, a typical number was in a range of approximately less than 1 million cells. The frequency of IFN-γ secreting cells as a fraction of CD4 or CD8 T cells is now presented in the supplemental information of the revised manuscript.

L387: It is not accurate to say that vaccination suppressed the induction of inflammatory cytokines, only that there were lower levels. It could be that these cytokines are produced in response to higher levels of virus in the lung, and that vaccinated animals were protected from viral replication, and therefor had lower levels.

Response: We agree to the point by this reviewer and revised the sentence of this description.

“In contrast, the adjuvanted (VSA-1, QS-21, Alum) sCal vaccination provided more effective control of lowering lung viral titers, which subsequently resulted in suppressing the induction of inflammatory cytokines (TNF-α, IL-6, IFN-γ, IL-1β) in the lung compared to the vaccine only group.” (page 14).

Figure 6D-E sCal+ alum data is not shown for these weight loss and viral titer protection graphs, yet it is stated in L389-391 that “These data suggest that VSA-1-adjuvanted split virus vaccination confers higher efficacy of protection against heterosubtypic virus than split virus only and Alum-adjuvanted vaccination.” Additionally, there was not a significant difference shown in the protection against viral lung titers between VSA-1+ sCal and unadjuvanted sCal. This is a major weakness of the paper, as a major claim is that this adjuvant enhances heterosubtypic protection, and this claim is not supported by the data. Further studies appear to be needed to support or reject this hypothesis, or the paper rewritten to limit conclusions.

Response: We agree to the concern of this point. Accordingly, we revised this description of the experimental outcomes, deleted the comparison with the alum-adjuvanted group, and limited the protective effects on preventing severe body weight loss after lethal challenge and increasing the survival rates (new data included in the revised manuscript). The efficacy of adjuvanted vaccination against preventing lung viral loads was low, in a range of 8 to 10 folds reduction, which is now moved to the supplementary information of the revised manuscript. (page 14).   

L394 and throughout the text: Induced is not the correct term, as no data is presented that any of the vaccines induced a T cell response independent of viral infection. All T cell data in the text is obtained from samples collected after challenge. And while these likely could be recall responses, it is also very possible that these formulations simply accelerated a primary T cell response to infection. Contrary to claims made throughout the paper (eg. L481-482), there is no data to demonstrate that any vaccines elicited a T cell response. While protection was dependent on administration of T cell depleting antibody treatments, there is no data to show that T cells were depleted by antibody treatment.

Response: We agree to this reviewer’s point on the T cell response data presented in this study. Accordingly, we revised the description on the T cell response data, to correctly reflect on how these data were obtained in this study throughout the revised manuscript.

Throughout the text, it is claimed that alum induced Th2 type responses, but Th2 cytokines were not measured.

Response: We revised and limited the description about the alum adjuvant effects, based on the experimental data presented.

  1. B. R. Park et al., Broad cross protection by recombinant live attenuated influenza H3N2 seasonal virus expressing conserved M2 extracellular domain in a chimeric hemagglutinin. Scientific reports 11, 4151 (2021).

Reviewer 2 Report

The authors described that VSA-1 played a significant role in enhancing IgG isotype and HAI antibodies after split influenza vaccine administration, conferring superior homologous protection over vaccine alone, and Alum- or QS-21-adjuvanted vaccination in C57BL/6 mice. VSA-1 was also highly effective in inducing humoral and cellular immune responses as well as cross-protection against H5N1 virus. Since VSA-1 was prepared from the inexpensive seeds of a widely available perennial vine and was more homogeneous and safer than QS-21, these findings are of interest and merit publication. However, the authors should discuss the results of in vivo protection efficacy test of immune sera or explain the aim of the test. Otherwise, the authors are suspected of being against animal welfare. The authors should also mention the structure of hemagglutinin in the introduction section to show the meaning of ELISA using HA-stalk.

Minor points:

1. Abbreviations should be written at the first appearance in the text.

2. Lines 114 and 119. Log103.7×EID50 was inadequate expression.

3. Lines 141 and 152. RPMI 1640 medium never contains FBS.

4. Line 179. CD4+ T.

5. Line 215. < 0.05 not ≤ 0.05.

Author Response

Responses to the comments from Reviewer 2

The authors described that VSA-1 played a significant role in enhancing IgG isotype and HAI antibodies after split influenza vaccine administration, conferring superior homologous protection over vaccine alone, and Alum- or QS-21-adjuvanted vaccination in C57BL/6 mice. VSA-1 was also highly effective in inducing humoral and cellular immune responses as well as cross-protection against H5N1 virus. Since VSA-1 was prepared from the inexpensive seeds of a widely available perennial vine and was more homogeneous and safer than QS-21, these findings are of interest and merit publication. However, the authors should discuss the results of in vivo protection efficacy test of immune sera or explain the aim of the test. Otherwise, the authors are suspected of being against animal welfare. The authors should also mention the structure of hemagglutinin in the introduction section to show the meaning of ELISA using HA-stalk.

Response: The authors appreciate for positive comments.

In response to this reviewer’s general comments, we revised manuscript accordingly. “Hemagglutinin (HA), a major antigenic target of vaccination, contains the HA1 immunodominant variable head domain and the relatively conserved HA2 stalk domain being a target of cross protective immunity (1).” (Introduction, page 1). “The aim of in vivo efficacy test is to determine the correlative role of adjuvanted-vaccine induced antisera in conferring homologous protection (Figs. 5E and F).” (Results, Page 11).

Minor points:

  1. Abbreviations should be written at the first appearance in the text.

Response: We made every effort in defining “Abbreviations” at the first appearance in the text of the revised manuscript.

  1. Lines 114 and 119. Log103.7×EID50was inadequate expression.

Response: We revised to describe the units appropriately:  every effort in defining “Abbreviations” at the first appearance in the text of the revised manuscript. “A/Cal H1N1 virus (3 × LD50, equivalent to 5x103 EID50),          heterosubtypic rgH5N1 virus (3 × LD50, equivalent to 2.5x103 EID50)”

  1. Lines 141 and 152. RPMI 1640 medium never contains FBS.

Response: We corrected this description on the culture media.

  1. Line 179. CD4+T.

Response: We corrected this: CD4 T cells

  1. Line 215. < 0.05 not ≤ 0.05.

Response: We corrected this

Round 2

Reviewer 1 Report

Overall, the main concerns in the first round of revision have been addressed, that conclusions must be carefully stated based on the data, and the authors have revised the text to that end. Additionally, the authors have included FACS plots to show readers how levels of ex vivo produced cytokines were measured. There are still a few minor concerns in the text.

L405 Based on the body weight loss trend in sCal only and sCal + VSA-1 groups, it is possible that the day 6 post infection time-point to assess lung viral titers might be an early time-point to see significant difference.

There is no way to make this determination, particularly in the results portion of the text. Without any data or references to suggest that clearance differences may be statistically visible on days 7, 8 etc, vs. day 6, the authors should consider leaving this for a point of discussion, or not including this statement.

L409 In contrast, the adjuvanted (VSA-1, QS-21, Alum) sCal vaccination provided more effective control of lowering lung viral titers, which subsequently resulted in suppressing the induction of inflammatory cytokines (TNF-α, IL-6, IFN-γ, IL-1β) in the lung compared to the vaccine only group.

We discussed the levels of cytokines in lungs after influenza virus challenge in the first round of revision because in the preliminary text, it was suggested that these adjuvant formulations were lowering cytokine production, and I pointed out that it “may” be possible that this is due to increased viral control. This was by no means to suggest that this is the primary mechanism which led to lower cytokine levels in the lungs. The authors should not take the discussion of this possibility to mean that they should present lower cytokine levels as a metric for increased viral control. This could be discussed, in the discussion portion of the text, but should not be presented as a result demonstrating vaccine induced viral control. Please revise this and other similar (L301) statements to a point of discussion.

The text could benefit from some editing to improve grammar and phrasing, but overall is readable.

Author Response

Overall, the main concerns in the first round of revision have been addressed, that conclusions must be carefully stated based on the data, and the authors have revised the text to that end. Additionally, the authors have included FACS plots to show readers how levels of ex vivo produced cytokines were measured. There are still a few minor concerns in the text.

L405 Based on the body weight loss trend in sCal only and sCal + VSA-1 groups, it is possible that the day 6 post infection time-point to assess lung viral titers might be an early time-point to see significant difference.

There is no way to make this determination, particularly in the results portion of the text. Without any data or references to suggest that clearance differences may be statistically visible on days 7, 8 etc, vs. day 6, the authors should consider leaving this for a point of discussion, or not including this statement.

Response: We agree with the above comments and have revised this whole paragraph by removing this sentence from the results section of the manuscript.

Based on the body weight loss trend in sCal only and sCal + VSA-1 groups, it is possible that the day 6 post infection time-point to assess lung viral titers might be an early time point to see a significant difference. High levels of TNF-α, IL-6, IFN-γ, and IL-1β cytokines were induced in the lung samples from naïve mice (Supplementary Figs. S5A-D) 6 days after infection. In contrast, the adjuvanted (VSA-1, QS-21) sCal vaccine groups showed lower levels of displayed provided more effective control in lowering lung viral titers, which subsequently resulted in suppressing the induction inflammatory cytokines (TNF-α, IL-6, IFN-γ, IL-1β) in the lung compared to the naïve mice with infection.”

L409  In contrast, the adjuvanted (VSA-1, QS-21, Alum) sCal vaccination provided more effective control of lowering lung viral titers, which subsequently resulted in suppressing the induction of inflammatory cytokines (TNF-α, IL-6, IFN-γ, IL-1β) in the lung compared to the vaccine only group.

We discussed the levels of cytokines in lungs after influenza virus challenge in the first round of revision because in the preliminary text, it was suggested that these adjuvant formulations were lowering cytokine production, and I pointed out that it “may” be possible that this is due to increased viral control. This was by no means to suggest that this is the primary mechanism which led to lower cytokine levels in the lungs. The authors should not take the discussion of this possibility to mean that they should present lower cytokine levels as a metric for increased viral control. This could be discussed, in the discussion portion of the text, but should not be presented as a result demonstrating vaccine induced viral control. Please revise this and other similar (L301) statements to a point of discussion.

 Response: We revised this sentence, by removing the phrase directly linking the lung viral titers and levels of inflammatory cytokines.

“Adjuvanted vaccine groups showed lower levels of lung viral titers, which resulted in reduced inflammatory cytokines (IL-6, IFN-γ, IL-1β) compared to the vaccine only or naïve mice after infection (Supplementary Figs. S2A-F).”

The text could benefit from some editing to improve grammar and phrasing, but overall is readable.

Response: We made some edits to further improve the readability of the manuscript.